# Are All Prognostic Stage IB Breast Cancers Equivalent? [note 1]

**DOI:** 10.3390/cancers16223830

**Published:** 2024-11-14

**Authors:** Stephanie M. Yoon, Shengyang Wu, Amanda Schwer, Scott Glaser, Todd DeWees, Jose G. Bazan

**Affiliations:** 1Department of Radiation Oncology, City of Hope National Medical Center, Duarte, CA 91010, USA; 2City of Hope Arcadia Radiation Oncology, Arcadia, CA 91007, USA; 3Department of Radiation Oncology, City of Hope Orange County Lennar Foundation Cancer Center, Irvine, CA 92618, USA

**Keywords:** breast cancer, staging, prognostic staging

## Abstract

Pathologic prognostic stage IB breast cancer is heterogeneous. It includes hormone-positive/HER2-negative (HR+/HER2-) breast cancers that have spread to nearby tissues or lymph nodes to small (<2 cm) early-stage triple-negative breast cancer (TNBC). However, prior studies demonstrated that disease involvement in regional lymph nodes is a key factor in prognosis and risk for disease recurrence. The discordance between anatomic and prognostic stages presents a challenge in recommending personalized treatments. Therefore, we studied women with extreme ranges of anatomic staging and biomarker status within this prognostic stage: locally advanced (HR+/HER2-; pT3N1 or pT1-3N2, grade 1–2) and node-negative early-stage TNBC (T1N0, grade 2–3) in the National Cancer Database to better understand survival differences between these groups. We found that patients with LA-HR+/HER2- BC have significantly worse survival compared to those with ES-TNBC, despite both being classified as pathologic prognostic stage IB cancers and receiving all appropriate treatments. Our study findings indicate a need to improve breast cancer prognostic staging for more accurate survival predictions.

## 1. Introduction

Breast cancer (BC) is a highly heterogeneous entity, and significant progress has been made in understanding its complex clinical and biological characteristics. Histopathologic grade and biomarker status have been highly related to prognosis [1,2,3,4,5,6,7]. It is now standard for clinicians to tailor therapeutic recommendations based on biologic subtype, thus ushering in an era of personalized therapy. For a long time, breast cancer was staged based on anatomic extent of disease, but the American Joint Committee on Cancer (AJCC) recently recognized the prognostic influence of histologic and biologic variants in its 8th edition cancer staging manual [8]. 

The new pathologic prognostic staging (PPS) for BC is now applied for all patients undergoing surgical resection as their initial treatment prior to receipt of radiation or systemic therapy. PPS integrates both anatomic disease extent and biological factors, including tumor grade, biomarker status (hormone receptor [HR] status that includes expression of estrogen receptors [ERs] and/or progesterone receptors [PRs] on the surface of tumor cells), human epidermal receptor 2 (HER2) amplification status, and multigene assay results. However, PPS IB remains heterogeneous and includes a wide range of patients from locally advanced anatomic pathologic stage IIIA/B (pT3N1 or pT1-3N2, G1–2) HR-positive/HER2-negative BC (LA-HR+/HER2- BC) to those with early-stage anatomic pathologic stage IA (T1N0, G2–3) triple-negative BC (ES-TNBC). This wide range of disease characteristics creates a discordance between anatomic disease extent and tumor biology. Generally, node-positive BC is linked to a higher risk of distant recurrences and poorer prognosis compared to node-negative BC [9,10,11]. The inclusion of LA-HR+/HER2- BC in the same prognostic stage as those with node-negative, small (under 2 cm) TNBC has raised concerns among patients. There is a dilemma as to which factor—tumor biology or anatomical disease extent—has greater importance in prognosis to guide treatment recommendations [12,13]. 

Scarce data are available as to whether biologic subtype or anatomic staging (AS) have greater prognostic impact on clinical outcomes in such discordant situations. The primary objective of this study is to evaluate the clinical outcomes of PPS IB breast cancer in greater detail. We focus on patients at the extremes of both AS and bioreceptor status within PPS IB, as defined by AJCC 8th edition staging manual. Specifically, we examined OS in patients with LA-HR+/HER2- BC and those with ES-TNBC. We hypothesize that OS will be worse for patients with LA-HR+/HER2- BC compared to those with ES-TNBC, despite both groups being classified as PPS IB.

## 2. Materials and Methods

This retrospective cohort study was reviewed and approved by the institutional review board at City of Hope National Medical Center with an exemption for informed consent requirements due to use and analysis of deidentified patient data. Data were sourced from the National Cancer Database (NCDB), a publicly available, hospital-based database collected over 1500 accredited facilities by the American College of Surgeons Commission on Cancer [14]. NCDB captures approximately 75% of newly diagnosed cancers in the United States.

We identified adult patients (≥18 years old) with surgically resected pT3N1 or pT1-3N2, grade 1–2 locally advanced HR+/HER2- BC (LA-HR+/HER2-) and those with T1N0, grade 2–3 ES-TNBC from 2004 to 2017. We excluded patients with unknown values for the following variables since these are necessary for AS and PPS: primary tumor stage, HR and HER2 status, tumor grade, and/or pathologic TNM staging data. To maintain consistency in treatments across the cohort, we also excluded missing treatment information, as well as HR+/HER2- patients treated with neoadjuvant therapy since most patients with luminal A tumors do not typically undergo this treatment and the prognostic pathologic stage may be influenced by neoadjuvant therapy. 

The study cohort was characterized using descriptive statistics. The primary endpoint was OS, which was estimated using the Kaplan–Meier method. OS was calculated from the date of diagnosis to the date of death or last follow-up. Log-rank test assessed for differences in OS between anatomic LA-HR+/HER2- BC and ES-TNBC (adjusting for baseline patient demographic characteristics). Survival analysis was performed in the entire cohort and in the subset of patients receiving all appropriate treatment based on AS: (1) radiation (RT), chemotherapy (CT), and hormone therapy for LA-HR+/HER2- BC and (2) CT or CT+RT for ES-TNBC treated with mastectomy or lumpectomy, respectively. We used multivariable Cox regression analysis to evaluate the association of BC subtype and AS with OS among the subset of patients who received all appropriate treatments. The primary predictor variable was anatomic BC stage: LA-HR+/HER2- BC or ES-TNBC. Covariables potentially impacting OS were included in the analysis: age (<40, 40–49, 50–59, 60–69, ≥70 years old), medical comorbidities (Charlson–Deyo Score 0 vs. ≥ 1, race/ethnicity (White, Black, Hispanic, Asian, Other, Unknown), surrogates for socioeconomic factors (insurance status [private, government, uninsured, unknown], median income [<USD 63,000, ≥USD 63,000, unknown], education [high-school diploma < 13.0%, ≥13.0%, unknown], and treatment facility type (academic, other, unknown). Further, given that the hazard ratio (HR) for recurrence over time appears non-proportional for TNBC but remains relatively constant for HR+/HER2- BC, we hypothesized that the HR for OS between the two groups could vary with time [15]. We implemented landmark analyses every 3 months, choosing the optimal landmark based on the time at which the adjusted hazard ratio for OS demonstrated a peak followed by a plateauing effect. We report hazard ratios with 95% confidence intervals (CIs) with *p* < 0.05 considered statistically significant. 

## 3. Results

We included a total of 45,818 patients with PPS IB treated between 2004 and 2017 that met the inclusion criteria: 17,359 patients had LA-HR+/HER2- BC and 28,459 had ES-TNBC. CONSORT diagram for included patients is found in Appendix A, and patient demographics are summarized in Table 1. Notably, 77.9% of patients were ≥ 50 years, and 82.6% had no associated comorbidities. Of patients with LA-HR+/HER2- BC, 13,064 (75.3%) had AS pT1-3N2 disease while the remaining had pT3N1 disease. Table 2 summarizes treatment delivery by AS groups. Approximately two-third of patients in each group received appropriate treatment based on AS: 11,533 (66.4%) patients with LA-HR+/HER2- BC and 19,512 (69.6%) with ES-TNBC.

Median follow-up for the entire cohort was 56.3 months (IQR = 35.8–80.9 months). Six-year OS rates were 86.1% (LA-HR+/HER2- BC) vs. 90.4% (ES-TNBC). Compared to patients with ES-TNBC, patients with LA-HR+/HER2- BC had a 63% relative increased risk of death after adjusting for all covariates (HR = 1.63; 95% CI 1.53–1.73; *p* < 0.0001) (Figure 1a). In addition, Appendix A shows that age ≥ 70 years having ≥ 1 medical comorbidity, ethnoracial category, insurance status, income, and facility type were all significantly associated with OS.

In the 31,045 patients that received appropriate treatment, the median follow-up was 58.7 months (IQR = 37.9–83.0 months). Six-year OS was 91.8% (LA-HR+/HER2- BC) vs. 93.3% (ES-TNBC). Patients with LA-HR+/HER2- BC had a 35% increased risk of death (HR = 1.35; 95% CI 1.24–1.48; *p* < 0.0001) compared to those with ES-TNBC after adjusting for all covariates (Figure 1b). Findings from Cox regression are summarized in Table 3. 

### Landmark Analysis

As demonstrated in Figure 1, absolute 5-year OS was worse amongst patients with LA-HR+/HER2- BC in the entire cohort (89.4% vs. 92.2%) and essentially identical in those receiving appropriate treatment (94.2% vs. 94.5%). As we hypothesized, Appendix A demonstrates the HR for OS between the LA-HR+/HER2- BC and ES-TNBC groups varied significantly as a function of time. At 54 months, there was a sharp increase in the HR. We therefore chose 54 months as the cut point for the landmark analysis. In the entire cohort, there were 24,057 patients (*N* = 14,966 ES-TNBC; *N* = 9091 HR+/HER2- BC) with a follow-up of ≥ 54 months (median = 79.5 months; IQR = 65.8–95.9 months) and a total of 1510 events. The adjusted HR for OS was 2.26 (95% CI = 2.04–2.50) with full results for the multivariable analysis in Appendix A. In the subgroup of patients who received all appropriate treatments, there were 17,112 patients (*N* = 10,591 ES-TNBC; *N* = 6521 HR+/HER2- BC) with ≥ 54 months of follow-up (median = 80 months, IQR = 66.0–96.3 months) and 751 events. Table 4 shows that the adjusted HR for OS was similar to that of the entire cohort with a value of 2.30 (95% CI = 1.99–2.66). The corresponding 7-year OS was 96.7% in patients with ES-TNBC vs. 93.2% in patients with LA-HR+/HER2- BC (Figure 2).

## 4. Discussion

In this study, we took the most extreme groups of patients that fall within prognostic stage IB breast cancer—those with anatomic pathologic stage IIIA-B (pT3N1 or pT1-3N1, grade 1–2 HR+/HER2- BC) and those with T1N0, G2–3 TNBC. Our goal was to evaluate OS and determine if clinical outcomes truly differ between these groups. We found that the OS was not the same between ES-TNBC (T1N0, grade 2–3) and LA-HR+/HER2- BC (pT3N1 or pT1-3N2, grade 1–2). Specifically, we found that patients with LA-HR+/HER2- BC have worse OS, and this OS difference persisted even when we restricted the analysis to patients who received appropriate oncologic treatment based on their AS of disease. These findings suggest that pathologic lymph node involvement should not be overlooked in HR+/HER2- BC. The current PPS should not include pT3N1 or pT1-3N2, grade 1–2 HR+/HER2- BC as PPS IB based on this evidence.

Breast cancer is a highly heterogeneous disease. Advances in genomic and molecular profiling have allowed clinicians to consider several other factors when tailoring treatment plans and predicting outcomes more accurately. Briefly, HR-positive BC generally has a more favorable prognosis compared to HR-negative BC or TNBC. This subtype has a slower growth trajectory and lower (though prolonged) incidence for relapse, which provide opportunities to intervene with effective treatments like endocrine therapies [6,16,17,18,19]. HER2-positive BC initially was associated with poor outcomes, but with routine use of anti-HER2-directed therapies, this subtype now has better prognosis [17,20,21,22]. TNBC is associated with worse prognosis due to its propensity for higher early disease relapse and lack of effective therapies leading to early death [4,5,23,24]. Histologic grade also has been demonstrated to have impact on BC prognosis [25,26]. Therefore, BC prognosis is no longer solely based on anatomic disease extent, and the AJCC 8th edition staging manual confirmed the value of biologic factors in predicting prognosis.

PPS has been externally validated across several institutions [13,21,27,28,29,30,31,32,33,34]. These studies have consistently shown improved discriminatory power over AS which confirmed its use in routine clinical practice. Weiss et al. compared disease-specific survival (DSS) according to AS and PPS across two cohorts of patients identified from a single institution and a large population database [27]. They reported that PPS provided more accurate DSS stratification compared to its associated AS (C-index = 0.737; *p* > 0.001) among the 3327 patients with stage I-III BC identified at the MD Anderson Cancer Center (MDACC) treated between 2007 and 2013. Similar results were found after assessing 54,727 patients with stage I-IV BC from 2005 to 2009 in the California Cancer Registry (C-index = 0.842; *p* < 0.001). However, most of these studies cross-validated PPS with the same AS. PPS upstaged 29.5% and 31.0% of patients in the MDACC and California Cancer registry cohorts, respectively, while it downstaged 28.1% and 20.6% of patients. Validation on non-metastatic patients identified in the surveillance, epidemiology, and end results (SEER) database also confirmed improved discriminatory ability in predicting BC-specific mortality, with C-statistic of 0.767 (95% CI = 0.759–0.776) with AS and 0.814 (95% CI = 0.807–0.822) with PS [21]. This study also performed stepwise pair comparisons between stages and found significant differences between stages (*p* < 0.05). Most other validation studies have compared PPS with its associated AS or within a specific BC subtype. 

However, PPS categorizes a wide range of AS from various molecular subtypes into the same prognostic stage. Currently, PPS IB encompasses a range of patients from early-stage anatomic clinical/pathologic stage IA TNBC to LA-HR+/HER2- BC. The discordance between AS, particularly for LA-HR+/HER2- BC, with prognostic stage presents a clinical challenge in recommending personalized treatments especially since prior studies demonstrated greater nodal involvement is a key determinant in prognosis and is significantly associated with an increased risk for disease recurrence [35,36]. In this study, we found that LA-HR+/HER2- BC has significantly worse OS compared to node-negative ES-TNBC despite both being classified as prognostic stage IB and accounting for treatments delivered. Worsened survival was more evident ~4–5 years after treatment. 

Two SEER 18 database studies have confirmed the improved performance of PPS over AS within LA-HR+/HER2- BC and TNBC patient populations. Wang et al. specifically compared PPS to AS among 10,053 patients with LABC in the SEER 18 database from 2010 to 2013 [32]. They found that 57.1% of cases with LA-HR+/HER2- BC were downstaged, while 60.4% of cases with grade 3 and 68.3% of cases with locally advanced TNBC were upstaged. PPS remains a significant independent prognostic factor for OS on multivariable analysis. When patients were stratified based on PPS, there was no significant difference in DSS for prognostic stage IB-IIIA but was significantly different compared to prognostic stage IIIB-IIIC. Specific pairwise comparison between prognostic stage IB to IIIA also did not have significantly different DSS. This study highlights that hormone receptor status in LABC may carry similar prognosis as early-stage BC [37]. Molecular status may have greater prognostic impact in predicting clinical outcomes for patients with discordant anatomic disease extent and biologic subtypes such as LA-HR+/HER2- BC. Another study led by Luo et al. validated PPS for patients in the SEER 18 database with TNBC treated from 2010 to 2015 [13]. They found that more than half of patients were upstaged based on PPS and no cases were downstaged. The performance of PPS outperformed AS among this patient population. Interestingly, they found that when different AS were distributed to the same prognostic stage, and there were significant differences in survival curves, which suggest that anatomic disease extent remains a relatively important factor to consider for TNBC. 

Anatomic extent of disease is an extremely important prognostic marker for patients to know, yet even in an era before prognostic staging was developed, data demonstrate that patients have poor understanding of the meaning of stage [38]. Patients need to know their stage of disease not only to guide prognosis but also to aid in treatment decision making as patients often express that the treatment they receive for their cancer should be in line with the severity of the disease [39]. We recognize PPS IB encompasses a wide range of AS; yet we purposely included patients with LA-HR+/HER2- breast cancer in this study because the most striking difference is present between AS (IIIA/B) and PPS (IB) for these patients. The 8th edition of the AJCC staging manual clearly states that, in patients that undergo upfront surgery and in countries where all prognostic variables are captured, PPS should be documented in the patient’s record. The staging manual also specifies that PPS is only accurate in patient’s that receive appropriate therapy for their anatomic extent of disease and biomarker status and even states that “lower stage does not denote the need for less treatment” [40]. In clinical practice, these subtle but important details need to be explained to patients, particularly those with pT3-1pN1-2 ER+/HER2-, grade 1–2 disease, such that they understand that while the medical record classifies them as having stage IB disease, they still need to complete all adjuvant therapy, including not only chemotherapy and radiation, but also endocrine therapy. We note that a slightly higher proportion of patients in our cohort with ES-TNBC (69%) completed all appropriate treatments compared to patients with LA-HR+/HER2- BC (66%), which potentially may impact clinical outcomes. However, evaluating OS as the primary endpoint may mitigate any inherent bias in this retrospective review. 

However, the key result from this study is that, even with appropriate treatment, the prognosis of patients with PPS IB LA HR+/HER2- BC appears to be worse compared to patients with PPS IB (T1c N0, G2–3) ES-TNBC. While the absolute differences in OS were small early in the study period, we undertook a landmark analysis with an a priori knowledge that recurrence rates are not proportional between TNBC and HR+/HER2- breast cancer. As demonstrated by Hilsenbeck et al., the risk of recurrence in patients with TNBC peaks sharply within the first 2–3 years of diagnosis and then declines to nearly 0, while the risk of recurrence in HR+/HER2- disease remains low but constant over time. Therefore, we see a divergence in the OS curves between the two groups we studied that begins at about 4 years and widens over time consistent with the natural history of HR+/HER2- disease. All of the patients with LA-HR+/HER2- breast cancer would have likely benefited from extended endocrine therapy as well as adjuvant cyclin-dependent kinase 4/6 inhibitors, though it is likely that many of the patients did not receive these treatments in the given study period [35,41,42]. With this in mind, it is possible that, if patients receive optimal adjuvant systemic therapy that is the standard of care today, the OS differences between LA HR+/HER2- BC and ES-TNBC may dissipate, but this can only be confirmed in future iterations of the NCDB that includes more patients treated in this manner.

To our knowledge, no studies have compared different AS and molecular subtype combinations within the same PPS. A Korean retrospective study performed prior to the establishment of PPS highlighted the prognostic importance of BC molecular subtypes by demonstrating that advanced AS (II-III) HR+/HER2- BC had better recurrence-free survival and OS compared to AS I, unfavorable histology (HER2+ or TN) BC [12]. Five-year recurrence-free survival was 99% vs 92%, respectively (*p* < 0.011), while five-year OS was 99% vs 96%, respectively (*p* = 0.03). While 5-year OS was similar to those found in our study, their conclusions contradicted our findings. This may be potentially due to several reasons including inclusion of AS II HR+/HER2- BC as an advanced stage, which could lead to better than expected OS compared to those with strictly AS III disease, and inclusion of HER2+ disease in AS I, unfavorable histology comparison group, which may also increase OS and/or recurrence-free survival rates compared to TNBC since over two-thirds of patients with HER2+ disease received anti-HER2-directed therapy, which has shown to improve clinical outcomes. We also recognize that the OS differences in our study were evident after 5 years. 

Overall, while evidence in the literature have demonstrated that LA-HR+/HER2- BC has very favorable clinical outcomes, our study highlights that anatomic disease extent remains important to consider in sub-population of patients with discordant AS and PPS [43]. Our findings suggest the current PPS needs further refinement and should not include LA-HR+/HER2- as PPS IB. Considering ~60–80% of all BC is HR+ with 90% of these staged as AS I-III, it is important to further clarify relevant biologic and anatomic factors that define disease at high risk for death or recurrence [44,45]. 

Our study has several limitations related to its retrospective nature and use of NCDB data for analysis [46]. First, the NCDB reports death from any cause; thus, it is possible our findings may differ if disease-specific deaths due to BC were considered. Second, the exact systemic therapy regimens and duration, particularly with respect to endocrine therapy, are not available in the NCDB. This study is hypothesis-generating, and our findings need further validation in independent cohorts of patients.

## 5. Conclusions

The AJCC 8th edition prognostic stage offers an advancement over anatomic staging alone by providing improved prognostic insights for breast cancer. However, our study indicates that classifying LA-HR+/HER2- BC (pT3N1 or pT1-3N2, G1–2) as pathologic prognostic stage IB inaccurately represents the actual prognosis for this group. A revised classification would provide patients with a more precise understanding of their expected overall survival, enabling more tailored clinical decisions. In the meantime, it is important for clinicians to convey these subtle but critical distinctions to patients and stress the importance of adhering to all recommended adjuvant therapy for optimal outcomes. 

## Figures and Tables

**Figure 1 cancers-16-03830-f001:**
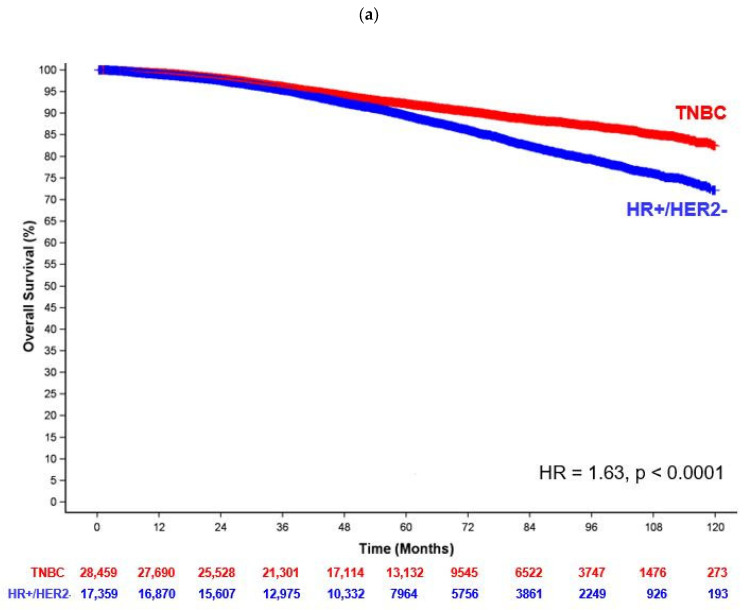
Kaplan–Meier curve for survival between locally advanced HR+/HER2- breast cancer (LA- HR+/HER2- BC) and early-stage triple-negative breast cancer (ES-TNBC) among (**a**) the entire cohort and (**b**) patients receiving all appropriate standard treatment. Among the entire cohort, patients with LA-HR+/HER2- BC had 63% relative increased risk of death compared to those with ES-TNBC after adjusting for all covariates (*p* < 0.0001). Among patients receiving all appropriate standard treatments, patients with LA-HR+/HER2- BC had a 35% relative increased risk of death compared to those with ES-TNBC after adjusting for all covariates (*p* < 0.0001).

**Figure 2 cancers-16-03830-f002:**
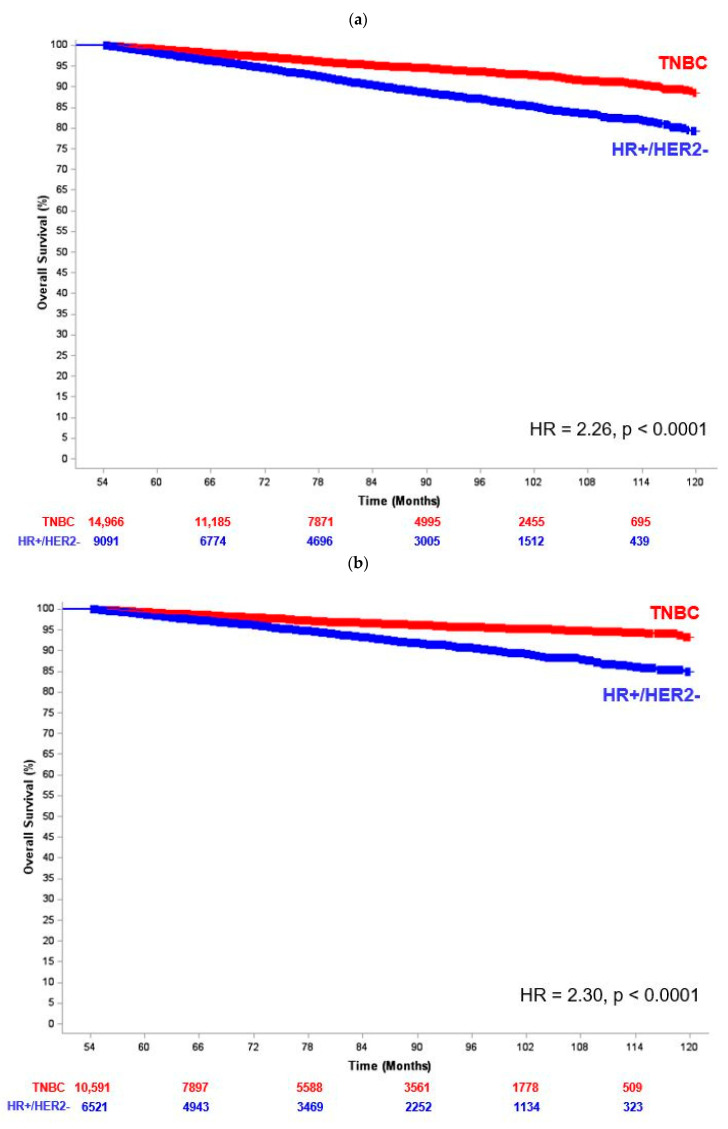
Kaplan–Meier curve for overall survival from landmark analysis (patients with ≥ 54 months of follow-up) between locally advanced HR+/HER2- breast cancer (LA- HR+/HER2- BC) and early-stage triple-negative breast cancer (ES-TNBC) among (**a**) the entire cohort and (**b**) patients receiving all appropriate standard treatment. Among the entire cohort, patients with LA-HR+/HER2- BC had 126% relative increased risk of death compared to those with ES-TNBC after adjusting for all covariates (*p* < 0.0001). Among patients receiving all appropriate standard treatment, patients with LA-HR+/HER2- BC had a 130% relative increased risk of death compared to those with ES-TNBC after adjusting for all covariates (*p* < 0.0001).

**Table 1 cancers-16-03830-t001:** Patient characteristics.

	Entire Cohort (*N* = 45,818)	LA-HR+/HER2- BC (*N* = 17,359)	ES-TNBC (*N* = 28,459)
Age			
<40 years	2224 (4.8%)	920 (5.3%)	1304 (4.6%)
40–49 years	7940 (17.3%)	3894 (22.4%)	4046 (14.2%)
50–59 years	12,304 (26.9%)	4609 (26.5%)	7695 (27.0%)
60–69 years	13,129 (28.7%)	4350 (25.1%)	8779 (30.9%)
≥70 years	10,221 (22.3%)	3586 (20.7%)	6635 (23.3%)
Comorbidities			
0	37,821 (82.6%)	14,398 (82.9%)	23,423 (82.3%)
1 or more	7997 (17.4%)	2961 (17.1%)	5036 (17.7%)
Treatment Facility			
Academic	13,040 (28.4%)	4736 (27.3%)	8304 (29.2%)
Other	30,554 (66.7%)	11,703 (67.4%)	18,851 (66.2%)
Unknown	2224 (4.9%)	920 (5.3%)	1304 (4.6%)
Race/Ethnicity			
White	33,343 (72.8%)	13,186 (76.0%)	20,157 (70.8%)
Black	6815 (14.9%)	1709 (9.8%)	5106 (17.9%)
Hispanic ^	2551 (5.6%)	1181 (6.8%)	1370 (4.8%)
Asian	1362 (2.9%)	608 (3.5%)	754 (2.7%)
Other	315 (0.7%)	140 (0.8%)	175 (0.6%)
Unknown Race or Ethnicity	1432 (3.1%)	535 (3.1%)	897 (3.2%)
Insurance Status			
Private	25,306 (55.2%)	9720 (56.0%)	15,586 (54.8%)
Government	19,236 (42.0%)	7045 (40.6%)	12,191 (42.8%)
Uninsured	792 (1.7%)	384 (2.2%)	408 (1.4%)
Unknown	484 (1.1%)	210 (1.2%)	274 (1.0%)
Median Income *			
≥USD 63,000/year	14,564 (31.8%)	55,89 (32.2%)	8975 (31.5%)
<USD 63,000/year	25,451 (55.6%)	9498 (54.7%)	15,953 (56.1%)
Unknown	5803 (12.6%)	2272 (13.1%)	3531 (12.4%)
No High School Diploma *			
≥13.0%	16,031 (35.0%)	5993 (34.5%)	10,038 (35.3%)
<13.0%	23,998 (52.4%)	9101 (52.4%)	14,897 (52.3%)
Unknown	5789 (12.6%)	2265 (13.1%)	3524 (12.4%)
Anatomic Stage			
IA, G2–3	28,459 (62.1%)	N/A	28,459 (100%)
IIIA (pT3N1), G1–2	4295 (9.4%)	4296 (24.7%)	N/A
IIIB (pT1-3N2), G1–2	13,064 (28.5%)	13,064 (75.3%)	N/A

* Measured by matching zip code of the patient recorded at time of diagnosis against files derived from the *American Community Survey* data; ^—all other racial categories not listed as “Hispanic” are specifically coded as non-Hispanic; abbreviations: LA-HR+/HER2- - locally-advanced hormone receptor-positive (including estrogen- and progesterone-receptor-positive) and HER2-negative; ES-TNBC—early-stage triple-negative breast cancer; G—grade; N/A—not available.

**Table 2 cancers-16-03830-t002:** Summary of treatments delivered by anatomic stage group.

Treatment Type	LA-HR+/HER2- BC (*N* = 17,359)	ES-TNBC (*N* = 28,459)
Breast Surgery		
Mastectomy	12,834 (73.9%)	7547 (26.5%)
Lumpectomy	4525 (26.1%)	20,912 (73.5%)
Radiation Therapy *		
Yes	13,932 (80.3%)	18,501 (88.5%)
No	2774 (16.0%)	1905 (9.1%)
Unknown/Other Site	653 (3.7%)	506 (2.4%)
Chemotherapy		
Yes	13,795 (79.5%)	20,724 (72.8%)
No	3437 (19.8%)	7624 (26.1%)
Unknown	127 (0.7%)	311 (1.1%)
Endocrine Therapy		
Yes	15,756 (90.8%)	N/A
No	1575 (9.0%)	N/A
Unknown	28 (0.2%)	N/A
All Appropriate Treatments **		
Yes	11,533 (66.4%)	19,512 (68.6%)
No	5826 (33.6%)	8947 (31.4%)

* For patients with ES-TNBC, reported only for lumpectomy patients; ** For LA-HR+/HER2- BC patients, this includes receipt of radiation therapy, chemotherapy, and endocrine therapy. For patients with ES-TNBC, this includes radiation and chemotherapy for patients that had lumpectomy and chemotherapy for patients that had mastectomy; abbreviations: LA-HR+/HER2- - locally-advanced hormone receptor-positive (including estrogen- and progesterone-receptor-positive) and HER2-negative; ES-TNBC—early-stage triple-negative breast cancer; N/A—not available

**Table 3 cancers-16-03830-t003:** Multivariable analysis of patients receiving all recommended treatments.

	HR (95% CI)	*p*-Value
LA-HR+/HER2- BC vs. ES-TNBC	1.35 (1.24–1.48)	<0.0001 *
Age		
<40 years	1.13 (0.90–1.42)	0.298
40–49 years	0.89 (0.77–1.02)	0.096
50–59 years	Reference	Reference
60–69 years	1.00 (0.88–1.13)	0.987
≥70 years	1.46 (1.26–1.70)	<0.001 *
1+ v 0 comorbidities	1.44 (1.29–1.61)	<0.0001 *
Race/Ethnicity		
White	Reference	Reference
Black	1.30 (1.15–1.47)	<0.001 *
Hispanic ^	0.83 (0.66–1.04)	0.102
Asian	0.63 (0.44–0.90)	0.010 *
Other	0.86 (0.46–1.61)	0.641
Unknown Race/Ethnicity	0.90 (0.70–1.16)	0.401
Insurance Status		
Private insurance	Reference	Reference
Government insurance	1.75 (1.57–1.95)	<0.001 *
No insurance	1.40 (1.01–1.94)	0.041 *
Unknown insurance	1.05 (0.64–1.72)	0.845
Median income		
≥USD 63 K/year	Reference	Reference
<USD 63 K/year	1.17 (1.04–1.32)	0.008 *
Unknown	1.60 (0.23–11.40)	0.638
Facility Type		
Academic	Reference	Reference
Non-academic	1.15 (1.03–1.27)	0.011 *
Unknown	-	-

LA-HR+/HER2- BC—locally advanced hormone-receptor-positive (including estrogen- and progesterone-receptor-positive) and HER2-negative breast cancer; ES-TNBC—early-stage triple-negative breast cancer; HR—hazard ratio; CI—confidence interval; * Statistically significant with *p* < 0.05; ^—all other racial categories not listed as “Hispanic” are specifically coded as non-Hispanic.

**Table 4 cancers-16-03830-t004:** Multivariable analysis of patients receiving all recommended treatments included in the landmark analysis with ≥ 54 months of follow-up).

	HR (95% CI)	*p*-Value
LA-HR+/HER2- BC vs. ES-TNBC	2.30 (1.99 – 2.66)	<0.0001 *
Age		
<40 years	1.17 (0.82–1.66)	0.395
40–49 years	0.77 (0.61–0.97)	0.025 *
50–59 years	Reference	Reference
60–69 years	1.03 (0.84–1.25)	0.788
≥70 years	1.73 (1.36–2.21)	<0.001 *
1+ v 0 comorbidities	1.58 (1.33–1.88)	<0.0001 *
Race/Ethnicity		
White	Reference	Reference
Black	1.27 (1.04–1.56)	<0.020 *
Hispanic ^	0.60 (0.40–0.91)	0.016 *
Asian	0.57 (0.32–1.01)	0.055
Other	0.46 (0.11–1.83)	0.268
Unknown Race/Ethnicity	0.82 (0.56–1.20)	0.316
Insurance Status		
Private insurance	Reference	Reference
Government insurance	1.54 (1.29–1.84)	<0.0001 *
No insurance	1.64 (1.03–2.62)	0.036 *
Unknown insurance	1.13 (0.56–2.29)	0.726
Median income		
≥USD 63 K/year	Reference	Reference
<USD 63 K/year	1.12 (0.92–1.34)	0.255
Unknown	00 (0.00–0.00)	0.959
Facility Type		
Academic	Reference	Reference
Non-academic	1.10 (0.93–1.29)	0.290
Unknown	-	-

LA-HR+/HER2- BC—Locally advanced hormone-receptor-positive (including estrogen- and progesterone-receptor-positive) and HER2-negative breast cancer; ES-TNBC—early-stage triple-negative breast cancer; HR—hazard ratio; CI—confidence interval; * Statistically significant with *p* < 0.05; ^—all racial categories not listed as “Hispanic” are specifically coded as non-Hispanic.

## Data Availability

Data cannot be provided by authors of this study as data were derived from the National Cancer Database (NCDB). Data can be requested directly from the NCDB.

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
