# Peer review of "Are All Prognostic Stage IB Breast Cancers Equivalent?â€"

_cancers, 2024, doi:10.3390/cancers16223830_

Round 1
Reviewer 1 Report (Previous Reviewer 1)
Comments and Suggestions for Authors
The Authors have addressed all the comments I had raised.
Author Response
We thank the reviewer for reviewing our revised manuscript.
Reviewer 2 Report (Previous Reviewer 2)
Comments and Suggestions for Authors
1. It is unclear in Figure 1 which groups are marked with which colors. Please add a decoding. The figure captions should be more detailed.
2. A similar remark regarding Figure 2. A more detailed description and decoding of the color scale is needed.
3. Is there information about the presence of mutations in the BRCA1/BRCA2 genes in the TNBC subgroup?
4. It still seems incorrect to me to compare prognostic group II (and subtype "T3, N1-2, M0, G2, HER2 Status: Negative, ER Status: Positive, PR Status: Positive" belongs to prognostic type II) and prognostic group IB in such an extreme version. The authors conclude that in the first case the prognosis is worse 86.1% (LA-HR+/HER2-) vs. 90.4% (ES-TNBC). But this is obvious anyway, especially considering that only stage T3N1M0 refers to primarily operable breast cancer. Other variants with lymph node involvement are considered primarily inoperable and the prognosis in these subgroups is obviously worse. The authors did not add a calculation of relative risks in case of different treatment regimens, as I asked at the previous stage of the review.
5. The authors conclude that the group "T3, N1-2, M0, G2, HER2 Status: Negative, ER Status: Positive, PR Status: Positive" is heterogeneous, but this conclusion is not new, so I doubt the relevance and significance of the work.
Author Response
We thank the reviewer for their feedback. We see the attachment for detailed point-by-point response to the reviewer's comments.

Reviewer 3 Report (Previous Reviewer 3)
Comments and Suggestions for Authors
The author's analytical method is sound, and the sample size is sufficient. However, the only issue is that it’s unclear who the target audience for this topic and research results are, as TNBC and ER+ patients have distinctly different treatment approaches. Even if it's known that LC-ER+ patients have a slightly worse prognosis than ES-TNBC patients, it’s not very clear what more can be done clinically or what physicians can offer to patients with this knowledge point.
Author Response
Feedback: The author's analytical method is sound, and the sample size is sufficient. However, the only issue is that it’s unclear who the target audience for this topic and research results are, as TNBC and ER+ patients have distinctly different treatment approaches. Even if it's known that LA-ER+ patients have a slightly worse prognosis than ES-TNBC patients, it’s not very clear what more can be done clinically or what physicians can offer to patients with this knowledge point.
Response: We thank the reviewer for this comment. Our study is intended for both physicians and patients. In response, we have revised the Conclusion section and parts of the Discussion section to more clearly convey that categorizing LA-HR+/HER2- breast cancer patients as pathologic prognostic stage IB inaccurately conveys the actual prognosis for these patients relative to ES-TNBC patients who are also categorized to this pathologic prognostic stage. Looking forward, we recommend that the upcoming versions of the prognostic staging system should reclassify LA-HR+/HER2- to provide more accurate prognostic information. In the meantime, it is essential for clinicians to stress the importance of completing all recommended adjuvant therapy (including hormone therapies, CDK4/6 inhibitors) to achieve optimal patient outcomes.
Round 2
Reviewer 2 Report (Previous Reviewer 2)
Comments and Suggestions for Authors
I have no comments/remarks on the manuscript.
Reviewer 3 Report (Previous Reviewer 3)
Comments and Suggestions for Authors
The author's statement is acceptable.
This manuscript is a resubmission of an earlier submission. The following is a list of the peer review reports and author responses from that submission.
Round 1
Reviewer 1 Report
Comments and Suggestions for Authors
In the present work, the Authors investigated the differences in overall survival (OS) in prognostic stage (PPS) IB breast cancer, comparing pT3N1/pT1-3N2 LA, HR +, HER2 - patients with early-stage pT1 TNBCs. They found that the formers were more likely to experience a severe clinical outcome, even when receiving the most appropriate treatment possible. The topic is indeed interesting, has it stresses out how pathological staging still carries remarkable prognostic implications even in favorable molecular subtypes (i.e. HR + tumors). Thus, this reviewer is with the Authors' claim of the need to redefine PPS groups, in order to underestimate the recurrence risk in truly high-risk HR + subjects. However, the Authors should address the following issues:
- The studied cohort spanned until 2017. As for HR + patients, would it be possible that they could have been given neoadjuvant chemotherapy if they had been diagnosed with breast cancer nowadays? Could have been chemotherapy proposed to luminal-A tumors as well, regardless of their Ki67 status?
- Within the HR subset, most patients were affected by N2 tumors (> 75%). May this finding represent a bias effacing the overall clinical outcomes of such a group?
- A slightly wider proportion of TNBC patients received the most appropriate treatments compared to HR + ones (69% vs 66%). Is it possible to speculate that such a difference, albeit limited, may have played a role in the clinical outcomes between the two cohorts?
Reviewer 2 Report
Comments and Suggestions for Authors
1. Provide in the Introduction section a scheme for classifying patients according to prognostic types 1A and 1B. By what principle is classification carried out?
2. Why is the anatomical stage and the nature of the treatment not taken into account in the multivariate analysis?
3. Figures 1 and 2 have unreadable inscriptions, please enlarge them.
4. According to AJCC v8 Stage IB includes: (T1, N0, M0, G1, HER2 Status: Negative, ER Status: Positive, PR Status: Negative); (T1, N0, M0, G1, HER2 Status: Negative, ER Status: Negative, PR Status: Positive); (T1, N0, M0, G2, HER2 Status: Positive, ER Status: Positive, PR Status: Negative); (T1, N0, M0, G2, HER2 Status: Positive, ER Status: Negative, PR Status: Any); (T1, N0, M0, G2, HER2 Status: Negative, ER Status: Negative, PR Status: Positive); (T1, N0, M0, G3, HER2 Status: Positive, ER Status: Negative, PR Status: Any); (T1, N0, M0, G3, HER2 Status: Negative, ER Status: Positive, PR Status: Positive); (T0-1, N1mi, M0, G1, HER2 Status: Negative, ER Status: Positive, PR Status: Negative); (T0-1, N1mi, M0, G1, HER2 Status: Negative, ER Status: Negative, PR Status: Positive); (T0-1, N1mi, M0, G2, HER2 Status: Positive, ER Status: Positive, PR Status: Negative); (T0-1, N1mi, M0, G2, HER2 Status: Positive, ER Status: Negative, PR Status: Any); (T0-1, N1mi, M0, G2, HER2 Status: Negative, ER Status: Negative, PR Status: Positive); (T0-1, N1mi, M0, G3, HER2 Status: Positive, ER Status: Negative, PR Status: Any); (T0-1, N1mi, M0, G3, HER2 Status: Negative, ER Status: Positive, PR Status: Positive); (T2, N0, M0, G1-3, HER2 Status: Positive, ER Status: Positive, PR Status: Positive); (T2, N0, M0, G1-2, HER2 Status: Negative, ER Status: Positive, PR Status: Positive); (T1, N1, M0, G1-3, HER2 Status: Positive, ER Status: Positive, PR Status: Positive); (T1, N1, M0, G1-2, HER2 Status: Negative, ER Status: Positive, PR Status: Positive); (T2, N1, M0, G1, HER2 Status: Negative, ER Status: Positive, PR Status: Positive); (T2, N1, M0, G2, HER2 Status: Positive, ER Status: Positive, PR Status: Positive); (T0-2, N2, M0, G1-2, HER2 Status: Positive, ER Status: Positive, PR Status: Positive); (T3, N1-2, M0, G1, HER2 Status: Positive, ER Status: Positive, PR Status: Positive); (T3, N1-2, M0, G2, HER2 Status: Positive, ER Status: Positive, PR Status: Positive). Why did the authors choose only IA, G2-3; IIIA (pT3N1), G1-2 and IIIB (pT1-3N2), G1-2? The subtype "T3, N1-2, M0, G2, HER2 Status: Negative, ER Status: Positive, PR Status: Positive" belongs to prognostic type II. The authors need to carefully and accurately explain these points (see point 1).
Reviewer 3 Report
Comments and Suggestions for Authors
The analytical method of the article is sound; however, the main issue lies in the inconsistency between the title and the research group settings within the article. I assume the author intends to discuss the prognostic survival of breast cancer patients who are classified under stage IB according to AJCC 8th but have different molecular subtypes. pT3N1 or > pT1N1 in LA-HR+/HER2- clearly exceeds the definition of stage IB. Even when considering PPS, breast cancer at pT3N1 cannot be classified as Stage IB. It would still be categorized into a higher clinical stage, typically Stage III. The author must re-filter the study population and redo the survival analysis to establish accurate results and conclusions.
Additionally, the data used in this study only extends up to 2017 and thus does not incorporate the concept of low HER2. In practice, some patients originally classified as TNBC might be HER2 1+ or 2+ patients who could potentially benefit from therapies that inhibit HER2 to improve survival.